# Sex Differences in Diabetes- and TGF-β1-Induced Renal Damage

**DOI:** 10.3390/cells9102236

**Published:** 2020-10-03

**Authors:** Nadja Ziller, Roland Kotolloshi, Mohsen Esmaeili, Marita Liebisch, Ralf Mrowka, Aria Baniahmad, Thomas Liehr, Gunter Wolf, Ivonne Loeffler

**Affiliations:** 1Department of Internal Medicine III, Jena University Hospital, Am Klinikum 1, D-07747 Jena, Germany; Nadja.ziller@med.uni-jena.de (N.Z.); Marita.liebisch@med.uni-jena.de (M.L.); Ralf.mrowka@med.uni-jena.de (R.M.); 2Institute of Human Genetics, Jena University Hospital, Am Klinikum 1, D-07747 Jena, Germany; Roland.Kotolloshi@med.uni-jena.de (R.K.); Mohsen.esmaeili@sickkids.ca (M.E.); Aria.baniahmad@med.uni-jena.de (A.B.); Thomas.liehr@med.uni-jena.de (T.L.)

**Keywords:** diabetic nephropathy, diabetes mellitus, kidney, tubulointerstitial fibrosis, sex differences, transforming growth factor beta 1, TGF-β1, testosterone, DHT, estradiol

## Abstract

While females are less affected by non-diabetic kidney diseases compared to males, available data on sex differences in diabetic nephropathy (DN) are controversial. Although there is evidence for an imbalance of sex hormones in diabetes and hormone-dependent mechanisms in transforming growth factor β1 (TGF-β1) signaling, causes and consequences are still incompletely understood. Here we investigated the influence of sex hormones and sex-specific gene signatures in diabetes- and TGF-β1-induced renal damage using various complementary approaches (a *db/db* diabetes mouse model, ex vivo experiments on murine renal tissue, and experiments with a proximal tubular cell line TKPTS). Our results show that: (i) diabetes affects sex hormone concentrations and renal expression of their receptors in a sex-specific manner; (ii) sex, sex hormones and diabetic conditions influence differences in expression of TGF-β1, its receptor and bone morphogenetic protein 7 (BMP7); (iii) the sex and sex hormones, in combination with variable TGF-β1 doses, determine the net outcome in TGF-β1-induced expression of connective tissue growth factor (CTGF), a profibrotic cytokine. Altogether, these results suggest complex crosstalk between sex hormones, sex-dependent expression pattern and profibrotic signals for the precise course of DN development. Our data may help to better understand previous contradictory findings regarding sex differences in DN.

## 1. Introduction

Type 2 diabetes mellitus (DM) is a worldwide epidemic disease with steadily increasing incidence due to changing lifestyles and increasing obesity [1].

Diabetic nephropathy (DN) is one of the major complications in patients with DM and the leading cause of terminal kidney failure in many countries [1,2]. Patients with type 1 diabetes mellitus (T1DM) and type 2 diabetes mellitus (T2DM) have a similarly high risk of developing DN (approximately 30–40%) [3,4].

DN is clinically characterized by albuminuria and progressive renal insufficiency. In early DN, the interstitium is expanded, and tubulointerstitial fibrosis (TIF) and tubular atrophy follow glomerular changes [5,6]. Features of TIF are myofibroblast accumulation, excessive deposition of extracellular matrix (ECM) and destruction of renal tubules [7,8]. There is complex crosstalk between fibroblasts, ECM proteins, and proximal tubular cells; proximal tubular injury in diabetes affects these interactions and contributes to TIF [9]. It is assumed that G2/M-arrested proximal epithelial cells upregulate expression of collagen and acquire a profibrotic phenotype by increased expression of cytokines such as transforming growth factor-beta 1 (TGF-β1) and connective tissue growth factor (CTGF), both being responsible for enhanced proliferation and collagen production of fibroblasts [10,11,12]. There is consensus that TGF-β1 is a principal inducer and modulator of a variety of pathophysiological processes in DN [2,13,14,15]. TGF-β1 initiates intracellular signaling by binding to receptor complexes which contain two distantly related transmembrane serine/threonine kinases. The cytokine binds first to the primary TGF-β receptor type 2 (TGFBR2), which is a constitutively active serine/threonine kinase that recruits receptor type 1 (TGFBR1) to the complex [13,16]. Subsequent transphosphorylation and activation of TGFBR1 by TGFBR2 allow the TGFBR1 kinase to phosphorylate selected Smads, which are inhibited by the so-called inhibitory Smads (Smad6 or Smad7) [13]. These receptor-activated Smads (R-Smads, e.g., Smad1/2/3/5/8) then form active complexes with the common Smad4, which translocate into the nucleus, where they regulate transcription of certain target genes [13].

CTGF is another profibrotic cytokine and has also been shown to be involved in DN, where it is strongly induced [14]. CTGF is an important downstream mediator of the profibrotic effects of TGF- β1, and TGF-β- and Smad-responsive elements in the CTGF promotor have been identified [17]. Downstream of a cascade of events induced by hyperglycemia, CTGF and TGF-β1 work in a coordinated manner to promote increased expression of extracellular matrix proteins (fibronectin and collagen types I, III, IV) [14]. Several antifibrotic and renoprotective agents have been shown to partially alleviate TGF-β -induced fibrosis and include bone morphogenic protein 7 (BMP7) [18]. Identified as an osteogenic factor and homodimeric member of the TGF-β superfamily, BMP7 is primarily expressed in kidney tubules and glomeruli and plays an important role in kidney development and the regulation of nephrogenesis [18,19]. In human and experimental DN, the renal cortical expression of BMP7 is progressively decreased [20,21]. BMP7 admission presents a potential therapy for DN, because in experimental settings this treatment leads to (i) reversed TGF-β -induced epithelial-to-mesenchymal transition (EMT), (ii) inhibited renal fibrosis, and (iii) reversed proteinuria to normal in a dose-dependent manner [18,20,22,23].

In non-diabetic kidney diseases, a strongly significant association between male sex and unfavorable renal outcome has been observed [24]. In most of the published literature on DN, it is essentially taken for granted that, as with non-diabetic kidney diseases, male sex is a risk factor for DN. However, a closer look at the literature reveals that not all studies support this assumption. Some studies revealed that men have a faster progression with DN than women, and other studies suggest DN progression to be faster in women (reviewed in [25]). Furthermore, onset and duration of DM, quality of glycemic control, puberty and menopause seem to play a major role in sex differences, too [25,26,27,28,29,30,31,32]. Thus, whether and to what extent sex hormones do play a role in DN are still a matter of controversial debate. Accumulating evidence suggests that diabetes is associated with an imbalance in sex hormone levels in both women and men, but it has not yet been conclusively clarified how the levels of testosterone and estrogen and their respective receptors related to disease progression in both sexes [25,26,27,28,33]. Women with T1DM have either reduced or unchanged estradiol levels compared to non-diabetic women, whereas testosterone levels reportedly are either elevated or similar to non-diabetic controls (reviewed in [25]). The findings in men with T1DM are similarly inconclusive: some studies report elevated levels of testosterone and estradiol, others report reduced levels or no changes in testosterone levels. In T2DM, testosterone is reduced in men and increased in women. Women with T2DM also have higher levels of estradiol, whereas men exhibit either elevated or no changes in estradiol levels (reviewed in [25]). Indeed there is some evidence of a paradigm shift away from the traditional simplistic assumption that testosterone is “bad” and estradiol is “good”. Yet, precise mechanisms by which sex hormones contribute to the pathophysiology of diabetic renal disease are poorly characterized. Sex hormones regulate many members of the BMP/TGF-β signaling pathways, which in turn strongly influence fibrotic processes in DN; other than this, very few studies have addressed the regulation of these pathways in a sex-dependent manner (reviewed in [34]).

We performed this study to investigate the extent to which sex hormones and/or the genetic repertoire play a role in sex differences in both diabetes- and TGF-β1-induced renal damage. Accordingly, in all experiments, a focus was placed on investigating interactions and synergies between all factors, i.e., sex, sex hormones, diabetes and TGF-β1.

## 2. Materials and Methods

### 2.1. Animals

All animal experiments were approved by the Local Ethics Committee (UKJ-17-024). The mouse model used in this study is the *db/db* mouse, which is the most widely used one for modeling DN in settings of T2DM [35]. *db/db* mice (BKS.Cg-m^+/+^Lepr db/J; C57BLKS/J background) were obtained from Charles River Laboratory (Brussels, Belgium). All animals were maintained in a pathogen-free facility with 12 h light/dark cycle and were reared on standard chow and water ad libitum.

For in vivo experiments, animals between 20 and 25 weeks of age were selected by genotype and divided into four groups: non-diabetic (BKS.Cg-m^−/−^Lepr db) males and females as well as diabetic (BKS.Cg-m^+/+^Lepr db) males and females.

Mice were placed in metabolic cages (Tecniplast, Buguggiate, Italy) to collect 24 h urine, body weights were measured, mice were euthanized, and cardiac blood was collected. Mouse blood was collected for determination of hematologic parameters using the pocH-100iV DIFF instrument (Sysmex, Norderstedt, Germany) and extraction of plasma for the detection of sex hormones. Mouse tail-vein blood glucose was measured with AlphaTrak (Abbott, Chicago, IL, USA).

### 2.2. Assessment of Renal Damage

Urinary albumin-to-creatinine ratio (ACR) was determined to quantify albuminuria. An ELISA specific for mouse albumin (Cell Trend, Luckenwalde, Germany) was used to measure urinary albumin excretion. Urinary creatinine was measured with a standard enzymatic assay (Cayman Chemicals, Ann Arbor, MI, USA).

For histological assessment of renal tubulointerstitial fibrosis, paraffin sections (thickness 2 μm) of the kidney were deparaffinized and exposed to heat-mediated antigen retrieval in citrate buffer (pH 6.0). Endogenous peroxidases were inactivated by pre-incubation with 3% H_2_O_2_ (Roth, Karlsruhe, Germany) for 10 min at room temperature. The primary antibodies rabbit anti-collagen I and rabbit anti-fibronectin (both Abcam, Cambridge, UK) were used. After incubation of the sections with peroxidase-labeled goat anti-rabbit IgG antibody (KPL, Gaithersburg, MD, USA), diaminobenzidine (DAB) was used as a chromogen (Peroxidase substrate Kit DAB, Vector Laboratories, Burlingame, CA, USA). For documentation and quantification, 10 non-overlapping images per section were acquired under standardized conditions by using AxioVision 4.8 software and the monochrome modus of the AxioCam HRc camera (both Zeiss, Jena, Germany).

### 2.3. Assessment of Sex Hormone Concentration in Blood Plasma

For plasma preparation, mouse blood samples were centrifuged for 20 minutes (min) at 1000 rpm at 4 °C, the supernatant was transferred in a new 1.5 mL tube and stored at −80 °C until the measurements. Sex hormone concentrations in plasma were determined using an ELISA (enzyme-linked immunosorbent assay) specific for mouse testosterone (Testosterone Saliva ELISA, TECAN, Männedorf, Switzerland), and mouse estradiol (Estradiol Parameter Assay Kit; R&D Systems, Minneapolis, MN, USA) and performed according to the manufacturer’s instructions.

### 2.4. Ex Vivo Model and Study Protocol

Ex vivo experiments with sex hormones were performed with kidneys from non-diabetic and diabetic mice of both sexes. In the ex vivo experiments with TGF-β1 stimulation and sex hormones, we only studied non-diabetic C57BLKS/J mice of both sexes.

Mice were euthanized, both kidneys removed and kidneys were decapsulated and rinsed with 2–3 mL ice-cold isolation medium (DMEM, Dulbecco′s modified eagle medium, 5.5 mM D-glucose, (Gibco, Thermo Fisher Scientific, Dreieich, Germany); supplemented with 0.5% Penicillin/Streptomycin (Pen/Strep) (Sigma Aldrich, Merck, Darmstadt, Germany; 10,000 U/mL). After removal of the medulla, the kidneys were minced into small pieces (1 mm^3^). All pieces of both kidneys per animal were equally distributed into a 12-well plate, filled with control medium (isolation medium supplemented with corresponding controls to stimulation compounds: citric acid, (pH = 3.0, 10 mM, solvent of TGF-β1); methanol (0.1%, solvent of dihydrotestosterone (DHT)); dimethyl sulfoxide (DMSO 0.1%, solvent of β-estradiol)), or stimulation medium (isolation medium supplemented with and/or 2.5 or 15 ng/mL TGF-β1 as well as with or without DHT (10^−6^ M = 2.9 ng/mL, Sigma Aldrich) and β-estradiol (estradiol, 10^−8^ M = 27.2 pg/mL, Sigma Aldrich)). Plates were further incubated at 37 °C and 5% CO_2_ for 24 h.

### 2.5. Cell Line and In Vitro Protocol

TKPTS cells (murine proximal tubular epithelial cell line) were cultured in DMEM (5.5 mM D-glucose; supplemented with 10% fetal calf serum (Gibco); 0.5% Pen/Strep, 10,000 U/mL, ~3.15 mg/500 mL medium Insulin solution, 10.5 mg/mL stock solution, Sigma Aldrich) at 37 °C, 5% CO_2_. In order to investigate the effects of TGF-β1 as well as impact of co-stimulation with sex hormones (DHT and estradiol), cells were seeded into 12-well plates (1 × 10^5^ cells/well), treated with starvation medium (DMEM, 5.5 mM glucose; without fetal calf serum; supplemented with 0.5% Pen/Strep, 10,000 U/mL) overnight, and further incubated with control medium (starvation medium supplemented with corresponding controls to stimulation compounds: citric acid, (pH = 3.0, 10 mM, solvent of TGF-β1); methanol (0.1%, solvent of DHT); DMSO (0.1%, solvent of β-estradiol)) or stimulation medium (starvation medium containing and/or 2.5 or 15 ng/mL TGF-β1 with or without DHT (10^−6^ M = 2.9 ng/mL), β-estradiol (10^−8^ M = 27.2 pg/mL)) for 24 h at 37 °C/5% CO_2_.

### 2.6. Immunofluorescence Staining

Kidneys were fixed with 10% neutral-buffered formalin and embedded in paraffin. Protein expression was analyzed on 3 µm kidney sections. The deparaffinized sections were subjected to heat-mediated antigen retrieval in citrate buffer (pH 6). After blocking with 5% BSA (bovine serum albumin, Roth), kidney sections were incubated overnight at 4 °C with the following primary antibodies: rabbit anti-androgen receptor (Abcam; 1:100, monoclonal), rabbit anti-estrogen receptor α (Biorbyt, Cambridge, UK; 1:100, polyclonal) and rabbit anti-TGFBR2 (Abcam; 1:100, polyclonal), and afterwards with the corresponding secondary antibody goat anti-rabbit Alexa 594 (Vector Laboratories; 1:500). Nuclei were counterstained with 4’,6-diamidino-2-phenylindole (DAPI, Sigma Aldrich). Nuclear or cytoplasmic expression was determined on 10 randomly selected, non-overlapping fields per mouse (three independent observers and investigators blinded to the origin of groups) with 5–10 mice per group. The expression was scored according to staining as follows: 0 = absent, 1 = light, 2 = mild, 3 = moderate, 4 = intense and 5 = strong. The overall mean scores were calculated based on individual values.

To localize androgen receptor (AR), estrogen receptor (ER) and TGFBR2 in the renal tissue, triple-immunofluorescence staining with a specific marker for proximal tubular cells (γ-glutamyltranspeptidase, GGT) and distal tubular cells (prominin-2, Prom2) was performed. To minimize background staining, the Vector^®^ TrueVIEW™ Autofluorescence Quenching Kit (Vector Laboratories) was used according to the manufacturer’s instructions. The three stainings were performed as described above and in sequence. First, the complete staining for AR/ER/TGFBR2 was performed, followed by the GGT staining (antibodies: primary goat anti-GGT (Santa Cruz Biotechnology, Dallas, TX, USA; 1:100, polyclonal); secondary horse anti-goat Alexa 488 (Vector Laboratories; 1:500)) and thirdly the Prom2 staining (antibodies: primary mouse anti-Prom2 (GeneTex, Irvine, CA, USA; 1:100, monoclonal); secondary donkey anti-mouse Cy5 (Invitrogen, Carlsbad, CA, USA; 1:500)).

Stainings were analyzed with the Zeiss Axio Imager.Z2 and Zen 2.5 blue software (Zeiss).

### 2.7. Characterization of TKPTS Cells

TKPTS cells were seeded in 8-well chamber slides. Confluent cells were fixed with ice-cold (−20 °C) methanol/acetone in a ratio of 1:1. After blocking with 5% BSA (Roth, Karlsruhe, Germany), they were incubated overnight at 4 °C with the following primary antibodies: rabbit anti-androgen receptor (Abcam; 1:200, monoclonal), rabbit anti-estrogen receptor (Biorbyt; 1:200, polyclonal), rabbit anti-TGFBR2 (Abcam; 1:200, polyclonal), mouse anti-CD31 (Abcam; 1:300, monoclonal), rabbit anti-E-cadherin (Cell signaling Technology, Leiden, Netherlands; 1:200, monoclonal) and mouse anti-Prom2 (GeneTex; 1:200, monoclonal), followed by the corresponding secondary antibody: goat anti-rabbit Alexa 594 (Vector Laboratories; 1:500), and horse anti-mouse Dylight 488-labeled antibody (Vector Laboratories; 1:500). Nuclei were counterstained with DAPI (Sigma Aldrich). Staining was analyzed with the Zeiss Axio Imager.Z2 and Zen 2.5 blue software (Zeiss).

### 2.8. Reverse Transcription and Real-Time PCR

Total RNA was isolated from either cells or kidney homogenates using the RNeasy Mini Kit (Qiagen, Hilden, Germany) or RNeasy Lipid Tissue Mini Kit (Qiagen), respectively. Possible DNA contamination was eliminated using the RNase-Free DNase Set (Qiagen), and depending on the RNA amount, 0.5–1 µg total RNA was reverse-transcribed using the Reverse Transcription System from Promega (Madison, WI, USA). The expression levels of genes were determined by semi-quantitative real-time PCR using LightCycler-FastStart DNA Master SYBR Green 1 (Roche Diagnostics, Mannheim, Germany) on a thermocycler (qTOWER, Analytik Jena, Jena, Germany). PCRs were carried out in a total volume of 20 µL containing 1–2 µL of cDNA and sense and antisense primers at a concentration of 0.25 µM each. To normalize for differences in the amount of cDNA in each sample, we performed amplification of hypoxanthine phospho-ribosyltransferase 1 (HPRT1) as housekeeping control. The amplification program included an initial denaturation step (5 min, 95 °C), 40 cycles of amplification (denaturation for 20 s at 95 °C), primer annealing (temperature see Table 1) for 20 s, extension for 20 s at 72 °C, an additional heating step to melt potential primer dimers for 15 s at 95 °C), and a melting curve program (denaturation for 15 s at 60 °C, cooling and holding for 15 s at 60 °C, and then heating at a speed of 0.1 °C/s to 95 °C). In the end, a cooling step of 2 min at 40 °C was performed. Table 1 shows the sequences and annealing temperatures of all primer pairs (purchased from TIB Molbiol, Berlin, Germany). Transcript levels were normalized to the expression of the housekeeping gene, the relative expression ratio was quantified by the ΔΔCT method and the non-diabetic males (in vivo experiments), or the control treatment of non-diabetic males (ex vivo experiments) or the control treatment of cells (in vitro experiments) were assigned as an arbitrary value of 1.

### 2.9. Statistical Analysis

The values given in this article are presented as the mean ± S.E.M. The graphs for ACR and the mRNA expression data from in vivo studies are shown as box and whisker plots with overlay of individual data points. Box and whisker plots are created using Sigma Plot (V12, Systat Software, Erkrath, Germany). The boundary of the box closest to zero indicates the 25th percentile, the line within the box marks the median, and the boundary of the box farthest from zero indicates the 75th percentile. Whiskers above and below the box indicate the 90th and 10th percentiles All results were analyzed using SPSS statistics (IBM company, Armonk, NY, USA). The in vivo effects were assessed using two-way ANOVA with sex (male or female) and diabetes (non-diabetic or diabetic) as the two factors. The ex vivo data were analyzed using three-way ANOVA with sex (male or female) and treatment (diabetes/TGF-β1) and hormones (DHT or estradiol) as the three factors. The data generated with TKPTS cells were analyzed using two-way ANOVA with treatment (TGF-β1) and hormones (DHT or estradiol) as the two factors. A *P* value of ≤ 0.05 was considered significant.

## 3. Results and Discussion

### 3.1. Diabetes Mellitus Causes Changes in Laboratory Parameters and Development of Diabetic Nephropathy

The *db/db* mouse model of leptin receptor deficiency is the most widely used model for DN in T2DM [35]. The animals develop obesity and elevated blood glucose levels from an age of approximately 6–8 weeks on. In this study, we investigated mice aged approximately 20–25 weeks, which corresponds to a medium to long diabetes duration in humans. As the clinical/laboratory data in Appendix A (Appendix A) show, DM has a significant influence on almost all parameters (except white and red blood cells). Sex alone significantly alters the number of platelets, MCHC and urine output. Interestingly, there is a significant interaction between the sexes and diabetes in body weights. While healthy females are, as expected, lighter in weight, they gain more weight with T2DM than males (Appendix A). Microalbuminuria is an early sign of DN, which is determined by the urinary ACR. Figure 1A shows that microalbuminuria (defined by ACR from >30 to <300 mg/g) is present in both sexes of diabetic mice, with no significant sex influence at this stage of DN (Figure 1A).

Accumulation of ECM results in a somewhat later phase of TIF, in that fibronectin and collagen IV are major ECM proteins that serve as a scaffold for the deposition of other proteins, such as collagen type I and III [17,43]. Immunohistochemical staining for fibronectin and collagen I shows the same picture as for ACR: a significant increase due to DM but no apparent sex differences (Figure 1B,C).

These clinical and pathohistological data illustrate the marked kidney damage caused by T2DM (mild albuminuria and TIF), yet they do not disclose any obvious significant sex differences, at least at this stage of DN.

### 3.2. Interaction between Sex and T2DM in Plasma Estradiol Levels and Renal Estrogen Receptor Expression

In order to further investigate sex differences in DN, we first addressed how T2DM influences circulating sex hormone concentrations and hormone receptor expression levels in renal tissue. We have addressed this issue in *db/db* mice (Figure 2). If we only focus on non-diabetic animals, as expected, we find that plasma testosterone levels as well as AR expression are higher in male animals and vice versa, estradiol levels in the plasma of female mice and their renal expression at ER are higher.

Although non-significant, T2DM causes a reduction in plasma testosterone levels in males to the level found in females (Figure 2A). This is consistent with several studies reporting that testosterone deficiency is common in men with diabetes [44,45,46,47,48,49,50]. Moreover, experimental models of diabetes (type 1 and 2) show reduced testosterone levels in diabetic males compared to non-diabetic controls of the same sex [33,51,52,53]. Due to the fact that approximately one-third of males with diabetes develop DN as a late consequence of DM, reduced testosterone levels in these subjects indicate that this sex hormone may not play a central role in the development of DN. On the contrary, it could even be shown that in streptozotocin (STZ)-diabetic male rats (a model for T1DM) a supplementation with DHT, the non-aromatizable and biologically more potent androgen, is partially renoprotective [54,55].

In contrast to the reduced testosterone level in diabetic males, female plasma testosterone levels do not appear to be different in non-diabetic and diabetic animals (Figure 2A). The sparse literature on testosterone levels in diabetic women or diabetic female animals tends to indicate elevated levels of total or free testosterone in DM [49,56,57,58], but two publications report the opposite [59,60].

In our study, in *db/db* mice, AR expression in the kidneys is unchanged from the previously described physiological conditions: high in males and lower in females (Figure 2C). Regarding the estradiol levels in T1DM and T2DM in humans and animals, the literature is relatively unanimous: there is an increase in estradiol in DM in male sex and there is a decrease in females [33,49,51,53,56,57,58,59,60,61,62]. Our data on circulating estradiol in healthy and diabetic mice of both sexes are consistent with these previously described changes (Figure 2B). Looking at individual group comparisons, estradiol and its receptor are upregulated in diabetic males, while a decrease occurs in diabetic females compared to healthy mice. This leads to equal levels of plasma estradiol and ER in diabetic males and females—sex differences are no longer observed (Figure 2B,D). There is a strong interaction between the effects of sex and DM on circulating estradiol levels and on renal ER expression. Including the fact that weight gain is significantly higher in diabetic females than in diabetic males (Appendix A), one would expect estradiol to be higher in diabetic obese females than in diabetic males, because fatty tissue is an additional source of estradiol. Apparently, the lowering effect of diabetes on female plasma estradiol concentrations is strong enough to eliminate sex significance, as shown in Figure 2B.

An interesting study shows that supplementation with estradiol in female STZ-induced diabetic rats as well as blocking of estradiol synthesis in male STZ-induced rats partially attenuated diabetes-associated renal injury [54,63]. This, together with our data on changes in estradiol levels, suggests that estradiol in diabetic females is partly renoprotective, but contributes to the development of DN in males.

Our data provide a complete picture of the imbalance of sex hormone levels and the expression of their receptors in both sexes in the presence of T2DM and reveal a sex–diabetes interaction for plasma estradiol as well as for ER expression, which is shown for the first time in our study.

### 3.3. Murine Tubular Cells Express the Receptors Required for Sex Hormone and TGF-β1 Signaling

Figure 2 shows the expression of sex hormone receptors in the renal cortex of male and female mice under physiological and pathophysiological conditions. To distinguish in which cell type of the tubular system (proximal or distal) the receptors are primarily expressed, co-immunofluorescences of AR or ER with specific markers for proximal (GGT) and distal (Prom2) tubule cells in murine kidney sections were performed. The stainings display that the AR is most strongly expressed in proximal tubules and the ER in distal tubules (Figure 3A,C). Expression studies of TGFBR2 show a similarly strong expression of the receptor in proximal and distal tubules (Figure 3E).

Since we were interested in hormone- and TGF-β1-dependent signaling in proximal tubule cells, we have extended our investigations to this cell type in an in vitro approach. The TKPTS cells were first characterized for their proximal epithelial phenotype and, as expected, showed their typical cobblestone-like arrangement, they are positive for the epithelial marker E-cadherin but negative for Prom2 (distal tubular cell marker) and CD31 (endothelial cell marker) (Appendix A). We prepared the chromosomes and determined the sex of the cells by fluorescence in situ hybridization (FISH) analysis (Appendix A). The metaphase analysis of the chromosomes revealed that the TKPTS cells had an almost normal set of chromosomes and application of a Y chromosome specific FISH-probe showed that the TKPTS cells are of male origin (Appendix A). Immunofluorescence staining of the androgen receptor in proximal TKPTS cells confirmed its nuclear expression (Figure 3B), as it can also be seen in tissue. Similarly, the proximal cells used in this study are positive for TGFBR2 (Figure 3F). This was previously shown for another murine proximal tubule cell line [16]. As explained above, the tissue staining of the ER suggests that it is expressed predominantly in distal tubule cells, which was also shown in another study [53], but staining from the proximal cell line clearly shows that this cell type can also express the ER (Figure 3D). To what extent a possible estradiol effect in the tissue is then also mediated more by the distal tubules than by the proximal ones cannot be answered here.

### 3.4. Sex and Diabetic Conditions Cause Changes in Expression Pattern of Molecular Markers for TIF in DN

The current paradigm of renal injury and TIF is that interstitial fibroblasts, which may take on the phenotypic appearance of activated myofibroblasts, are the major source of the expanded ECM [64,65]. Complex crosstalk between the different cell types in the tubular system and interstitial space is described in the literature [9]. For example, there is evidence for reciprocal paracrine activation of proximal tubular cells and fibroblasts: proximal tubular cells release fibrogenic signals to cortical fibroblasts, and vice versa renal fibroblasts can modulate proximal tubule cell growth and transport [9].

The pathophysiology of TIF is divided into four overlapping phases: (1) the cellular activation and injury phase, (2) the fibrogenic signaling phase, (3) the fibrogenic phase, and (4) the destructive phase [43]. In the initial priming phase, injury to the tubular cells within the kidney results in the formation of local inflammation with a release of proinflammatory and injurious molecules [65]. The second phase is characterized by the production of fibrosis-promoting factors, and in the third phase, the ECM production increases as well as matrix degradation decreases [43]. The excessive accumulation of ECM (e.g., fibronectin and collagen I) results in the fourth phase, in that the number of intact nephrons progressively declines resulting in a continuous reduction in glomerular filtration [43].

In our animals, at this stage of DN, no significant sex differences in diabetes-induced albuminuria and morphologically assessable TIF (accumulation of ECM) were manifested. Therefore, we were interested in the TIF on the molecular level and the expression profile of markers of its activation and fibrogenic signaling phase (Figure 4).

We investigated the renal mRNA expression of the acute phase protein SAA (serum amyloid A) (Figure 4A) because although SAA is primarily a plasma marker for chronic inflammation, it is also produced locally at inflammation sites in addition to its dissemination by systemic circulation. For example, all SAA isoforms in renal tissue of humans and mice with diabetic kidney disease are highly upregulated in T2DM, both in the glomerulus and tubulointerstitium [66].

Snail1, whose mRNA expression we also studied (Figure 4B), is a profibrotic transcription factor, that is activated by the TGF-β/Smad3 signaling pathway but also TGF-β-independently [67].

The early fibrosis marker CTGF (Figure 4C) has also TGF-β-dependent as well as -independent effects to enhance renal fibrosis [12]. CTGF lacks a dedicated receptor but appears to interact with other proteins to promote a profibrotic environment. CTGF is a necessary cofactor for TGF-β signaling via direct interactions with the TGF-β receptor, downregulation of Smad7 activity, and inhibition of BMP7 [68]. CTGF can be found in both tubular epithelial cells and fibroblasts in normal kidneys, but upregulation occurs after exposure to injurious stimuli [68].

The expression of TGF-β1 (Figure 4D) itself, the ratio of TGF-β expression to its antagonist BMP7 (Figure 4E) and the protein expression of TGFBR2 (Figure 4F), to which TGF-β binds directly, are of course also part of the investigations of the fibrotic signaling phase of TIF.

Our expression data show a significant influence of sex on the expression of CTGF, TGF-β1, the relationship between TGF-β1 and BMP7 expression and the TGFBR2 protein level. Diabetes, on the other hand, significantly influences the expression of SAA, CTGF, the TGF-β1/BMP7 ratio and TGFBR2 expression. A significant effect interaction between sex and diabetes is shown in the mRNA expression of SAA, although no significant single sex effect occurs (Figure 4A). Interestingly, there is also a very strong sex–diabetes interaction in the expression of Snail1, although here, based on the two-way ANOVA significance analysis, neither sex alone nor diabetes alone causes a significant major effect (Figure 4B). When the expression pattern of SAA, Snail1 and CTGF is analyzed in intergroup comparisons (one-way), some similarities become apparent. Firstly, the basal expression is significantly higher in female healthy kidneys compared to healthy males. Secondly, the diabetes-induced upregulation as in males is fewer or no longer detectable in females. Thirdly, as a result of the first two mentioned, no sex differences in diabetic mice are detectable (Figure 4A–C). This could provide an explanation for the conflicting literature data, since single comparisons of diabetic males and females or the influence of diabetes in only one sex may lead to wrong conclusions if the basal data are not known or not included.

It is widely documented and generally accepted that the potent profibrotic cytokine TGF-β1 and its receptor TGFBR2 are elevated in T1DM and T2DM [2,69]. In the current study, an increase in TGF-β1 levels was detected in diabetic males only, and did not reach significance (*P* = 0.07) (Figure 4D). However, this does not necessarily contradict earlier findings, as other studies also show a comparably weak increase in TGF-β1 levels, e.g., in diabetic male rats [33,51]. Importantly, since our data only document a snapshot of the amount of TGF-β1 mRNA, future studies will need to resolve TGF-β1 levels over a broader period of time to determine the extent of TGF-β1 over-production in the diabetic kidney.

Other animal studies have also revealed a very interesting relationship between the sexes and the basal TGF-β1 system in the kidney [70]. It could be shown that before puberty there are no significant differences in active TGF-β1 in both sexes, but after puberty an up to 3-fold increase in TGF-β1 production is observed in females, whereas in males the activation of latent TGF-β1 is more prominent [70]. This suggests that after puberty in male animals TGF-β1 activation becomes more efficient and the female animals compensate their reduced activation efficiency by increasing total TGF-β1 levels [70]. Exactly this fact, that the basal expression of TGF-β1 is significantly increased in healthy females compared to non-diabetic males, is also found in our studies (Figure 4D comparison: non-diabetic males and females). This may also explain the already discussed basal expression patterns of Snail1 and CTGF, both downstream targets of TGF-β1, in non-diabetic animals. In any case, starting from basal high TGF-β1 levels in healthy females, there is no further increase in the diabetic female kidney, but the physiological sex difference is maintained under pathophysiological conditions (Figure 4D comparison: diabetic male and female). Some reports have demonstrated that estradiol, at least in osteoblasts, osteoclasts and bone marrow-derived mesenchymal cells, induces TGF-β1 expression [37,71]. Nevertheless, with the data shown here, it cannot be concluded with certainty that the increased estradiol is responsible for the rise in TGF-β1 in diabetic males. The higher TGF-β1 expression in healthy female kidneys supports the assumption that TGF-β1 is induced by estradiol, but in the female kidney under diabetic conditions a different mechanism must be underlying. Firstly, estradiol levels are lowered by T2DM in females (Figure 2B) without simultaneous detection of a decreased TGF-β1 expression in the same animals and secondly, it could be shown for exactly this T2DM model that estradiol treatment of diabetic *db/db* mice even lowers TGF-β1 expression compared to untreated *db/db* mice [72]. Again, the data suggest that the possible renoprotective effect of estradiol in diabetic females disappears by lowering estradiol levels, but estradiol levels in diabetic males correlate with increase in profibrotic TGF-β1.

It is unlikely that the male sex hormone testosterone is responsible for TGF-β1 induction in diabetic males, since testosterone levels are reduced in these same animals compared to non-diabetic males (Figure 2A). Interestingly, testosterone even seems to have an inhibitory effect on TGF-β1 production, since an orchiectomy significantly increases TGF-β1 levels in non-diabetic males as well as enhances the diabetes-induced TGF-β1 expression in male kidneys [33,51,55]. This would not actually be expected, but rather a renoprotective effect of lowering testosterone levels by orchiectomy, when the male sex is a risk factor for DN whose main inducer is TGF-β1. Orchiectomized diabetic animals with low replacement testosterone therapy showed lower TGF-β1 levels than castrated diabetic males without testosterone, but diabetic males with high-dose testosterone therapy after orchiectomy showed higher TGF-β1 levels than castrated diabetic males without testosterone [55]. This study by Xu et al. was only performed with male animals [55], so the question remains unanswered whether female diabetic animals with altered DHT levels would have reacted similarly or differently with renal TGF-β1 expressions.

Looking at the ratio of expressions of TGF-β1 and its antagonist BMP7 (Figure 4E), the significant increase in female non-diabetic as well as diabetic kidneys versus the corresponding male study groups remains, but there is an additional significant increase in the TGF-β1/BMP7 ratio in female animals under diabetic conditions, which is also reflected in the highly significant sex–diabetes interaction. This means that, although TGF-β1 expression in females is not significantly altered at this stage, the diabetes-induced pathological reduction in the anti-fibrotic BMP7 is increased, leading to a shift towards profibrotic signals (Figure 4E). We assume that not only the levels of TGF-β1 alone are decisive for the progression of DN, but that the balance or imbalance between profibrotic and anti-fibrotic signaling, i.e., the TGF-β1/BMP7 ratio, is also important. Sexual differences in diabetic animals are also found in TGFBR2 expression, even at the protein level, in that diabetic females have more TGFBR2 than diabetic males (Figure 4F). The upregulation of TGFBR2 in diabetic animals is likely induced by albumin. In a previous work, we could show that albumin in a concentration found in proteinuria leads to transcriptional induction of TGFBR2 in cultured proximal tubular cells, which was reflected in increased TGFBR2 protein expression and was also demonstrated by increased binding of TGF-β1 to cell surfaces [16]. Since TGFBR2 is primarily engaged in the initial binding of TGF-β1, an increased receptor expression results in amplification of the TGF-β1 effects on tubular cells [16]. This would support our hypothesis that in kidneys of diabetic females, the TIF progresses faster or more strongly.

In summary, our data show no significant sex differences in renal function and fibrosis at the time of investigation. However, sex differences in expression levels of molecular markers and inducers of TIF could be found. This suggests that sex is less important in development of DN than in its progression and that possible sex differences in DN are related to the TGF-β1 system.

### 3.5. Sex Hormones Directly Influence the Expression of TGF-β1 and BMP7 in Cortical Tissue from Mouse Kidney

To address the question of whether the sex hormones have a direct influence on the expression of the main mediators of DN, we used an ex vivo approach, which allowed us to stimulate kidney tissue from non-diabetic and diabetic mice of both sexes with sex hormones and to determine the mRNA expression of CTGF, TGF-β1 and BMP7 (Figure 5A–C).

The renal tissue of each animal was divided in such a way that in an experimental duplicate, the tissue could be stimulated with control (DMSO), DHT and β-estradiol (both in physiological concentrations; 2.9 ng/mL DHT and 27.2 pg/mL β-estradiol; see also Figure 2A,B). After 24 h in this culture medium, no expression differences for CTGF were detectable neither between the individual interventions nor between the sexes or the diabetes phenotype (Figure 5A). In contrast, there are some differences in the proportion of TGF-β1 mRNA in the same total RNA (Figure 5B). The TGF-β1 levels in renal tissue from non-diabetic females are at least two times higher than those from non-diabetic males, regardless of the intervention, confirming the finding shown in Figure 4D. This difference is reflected in the ratio between TGF-β1 and BMP7 and is even slightly increased since BMP7 is less expressed in non-diabetic females than in healthy male kidneys (ratio in Figure 5C; BMP7 not shown). The increased TGF-β1 expression in ex vivo cultured female kidney tissue compared to the male under physiological conditions is no longer detectable in kidney tissue from diabetic females compared to diabetic males.

Our data also suggest that DHT affects TGF-β1 expression differently depending on the diabetes phenotype in male kidney tissue: DHT decreases TGF-β1 mRNA levels in non-diabetic tissues and increases them in diabetic tissues compared to DMSO control in the same males. Other studies also show that both in vitro and in vivo TGF expression can be differentially influenced by DHT, which in these studies was dependent on the cell type or DHT concentration [34,55,73]. In kidney tissue from female mice, DHT has minimal or inhibitory effects on TGF-β1 mRNA expression compared to DMSO control, regardless of whether the donor females were diabetic or non-diabetic (Figure 5B). Although an increase in TGF-β1 due to diabetes in kidney tissue is no longer detectable after 24 h ex vivo cultivation in the control medium, the diabetes phenotype has an increasing (male sex) or decreasing (female sex) effect on the mRNA of TGF-β1 depending on the sex in combination with DHT (Figure 5B). These described relationships are also expressed in the significance analysis: (1) significant main effects of sex and diabetes, (2) a significant interaction of the sex effect with the diabetes phenotype, (3) a significant interaction between the diabetes phenotype and DHT intervention and (4) a triple interaction of the factors sex, diabetes and DHT (Figure 5B, upper table).

Figure 5B also shows the influence of β-estradiol on TGF-β1 expression. In the non-diabetic phenotype of kidney tissue from both sexes, an inhibitory effect of the female sex hormone on TGF-β1 is shown, whereas estradiol does not cause any altered expression in the diabetic phenotype of female and male kidney tissue. In contrast to the TGF-β1 increasing or decreasing effect of DHT treatment of kidney tissue from diabetic males or diabetic females compared to kidney tissue from non-diabetic animals of the corresponding sex, estradiol-treated male and female renal tissue pieces show no differences in TGF-β1 expression depending on the diabetes phenotype (Figure 5B). In the statistical analyses, no significant triple interaction of the factors sex, diabetes and sex hormone was found in the case of estradiol, apart from the main effects which are significant (Figure 5B, lower table). That the most important effect of estradiol seems to be to disrupt TGF-β1 signaling, rather than to regulate its expression, is shown by an animal study in which in female mice overexpressing TGF-β1, treatment with estradiol ameliorated TGF-β1-induced progressive kidney disease without decreasing TGF-β1-expression [72].

The expression pattern of the TGF-β1/BMP7 ratio rarely differs from the pattern of TGF-β1 expression (Figure 5C vs. Figure 5B), except that, as mentioned above, the differences in basal expression between females and males are more pronounced and that at least β-estradiol treatment of kidney tissue from diabetic females tends to lower BMP7 expression, resulting in a higher ratio and a significant triple interaction between sex, diabetes and estradiol.

In summary, our ex vivo approach shows that renal tissue in the presence of DHT or β-estradiol exhibits an altered expression of TGF-β1 and BMP7. Here, sex, diabetes phenotype and hormones act alone or in combination on mRNA expression. In the pathological context DHT tends to influence TGF-β1 expression and estradiol tends to influence BMP7 expression. This indicates an interaction of the two sex hormones, which determines the relationship between the profibrotic TGF-β1 and the antifibrotic BMP7 differently depending on the sex. Although the antagonistic nature of TGF-β1 and BMP signaling in the same tissue has been directly observed in some models of organ fibrosis, the regulation of these signaling pathways in the individual sex has not yet been addressed and is shown here for the first time.

### 3.6. In the Presence of Sex Hormones, the TGF-β1-Induced CTGF Expression Is Differentially Regulated in Both Sexes and Additionally Inversed Depending on TGF-β1 Concentration

While Figure 5B shows the potential influence of the sex hormones DHT and β-estradiol on TGF-β1 expression depending on sex and diabetes phenotype, we investigated in a next step, a possible effect of the hormones on TGF-β1 signaling. Further, in an ex vivo approach, we stimulated renal tissue, this time exclusively from non-diabetic females and males, with low and high concentrations of TGF-β1 in the absence and presence of exogenous sex DHT or β-estradiol. The relative mRNA expression of CTGF as the main downstream mediator of TGF-β1 served as readout for TGF-β1 signaling.

TGF-β1 stimulates CTGF expression at low and high doses in the absence of hormones (Figure 6). While the different concentrations of TGF-β1 in the same sex do not seem to have any influence on the strength of CTGF expression, there are clear gender differences in the way that TGF-β1, no matter what the concentration, induces higher CTGF levels in female renal tissue than in male tissue.

Interestingly, DHT shows opposite effects depending on the sex. In the presence of the low concentration of TGF-β1, DHT has no or rather a lowering effect on TGF-β1-induced CTGF expression in male renal tissue, but an increasing effect in female renal tissue (Figure 6A). In the presence of high dose TGF-β1 it is the other way round: DHT enhances the TGF-β1-induced CTGF-mRNA in male tissue and has no or rather inhibitory effect in female renal tissue (Figure 6B).

Estradiol lowers the low dose TGF-β1-induced CTGF expression in the renal tissue of female mice (Figure 6A), but has no further reproducible effect on the TGF-β1-induced CTGF mRNA either in the male tissue or at the high TGF-β1 concentration (Figure 6A see only males and Figure 6B).

Comparing the relative strength of TGF-β1-induced CTGF expression between the two sexes, the greatest gender difference can be observed when (1) the kidney tissue is stimulated with TGF-β1 and no sex hormones are involved (females are stronger), (2) the kidney tissue is stimulated with low concentrated TGF-β1 and DHT, and (3) the kidney tissue is stimulated with higher concentrated TGF-β1 and estradiol. In any case, CTGF is then more strongly expressed in renal tissue from females.

What is also striking is that the expression patterns of TGF-β1 mRNA, which is found in the diabetes phenotype (Figure 5B), and CTGF mRNA, which is obtained by stimulation with high TGF-β1 concentrations (Figure 6B), are very similar. One could speculate that TGF-β1 possibly released by the diabetic kidney tissue was present in the stimulation medium in similar concentrations, but this has not been investigated.

### 3.7. The Opposite Effects of Sex Hormones in Dependency of TGF-β1-Concentration Is Confirmed in TKPTS Cells

The investigations from both the in vivo and ex vivo approaches are based on total RNA from cortical tissue. In addition to the proximal tubule cells, the tissue naturally contains cells of the glomeruli, endothelium and distal tubules, which contribute to the expression profile shown. However, since the proximal tubule cells seem to play the predominant role in the development and progression of TIF [74], we have investigated in a controlled in vitro approach how TKPTS cells respond to TGF-β1 stimulation in combination with sex hormones in their CTGF expression.

Figure 7 shows the relative CTGF mRNA levels after 24 h stimulation with low dose TGF-β1 (Figure 7A) and high dose (Figure 7B). Under these controlled conditions, a TGF-β1 dose effect is shown in the form that the higher TGF-β1 concentration induces CTGF expression most strongly (comparison of TGF-β1 minus DHT/β-estradiol in Figure 7A vs. Figure 7B), but this is also shown a long time ago [75,76]. Much more interesting is that, as in the ex vivo approach, an opposite effect of DHT is obvious, which is TGF-β1 dose-dependent: DHT enhances low dose TGF-β1-stimulated CTGF expression and lowers high dose induced CTGF levels almost to basal levels. Estradiol shows the same effect as DHT in TGF-β1-treated TKPTS cells, again qualitatively distinct depending on the TGF-β1 concentration: Estradiol increased CTGF expression induced by low dose TGF-β1 and decreased CTGF levels triggered by high-dose TGF-β1 (Figure 7A,B).

### 3.8. TGF-β1-Regulated TGFBR1 and TGFBR2 Expression Is Influenced by Sex Hormones

The data in Figure 6 and Figure 7 show that the sex hormones influence TGF-β1 signaling, resulting in reduced or increased CTGF mRNA level. To better understand the role of the sex hormones in the TGF-β1 system, we investigated whether DHT and estradiol affect the TGF-β1-induced expression of the receptors TGFBR1 and TGFBR2 (Figure 8).

TGFBR1 is induced in TKPTS cells by TGF-β1 in a concentration-dependent manner (low dose 2-fold, high dose 4-fold), which is enhanced by DHT at the low dose and inhibited at the high dose (Figure 8A). Estradiol shows a lowering effect on TGFBR1 expression in both stimulation concentrations of TGF-β1 (Figure 8A). TGF-β1 also has a concentration-dependent but opposite effect on TGFBR2 expression: increasing at low doses and decreasing at high doses (Figure 8B). The addition of DHT has a strong lowering effect on both the low TGF-β1 concentration-induced and the already reduced expression of TGFBR2 at the high TGF concentration. Estradiol has no effect on either the increasing effect of low TGF-β1 concentration or the decreasing effect of high TGF-β1 concentration (Figure 8B).

In addition, the possibility of a receptor threshold model for determining the specificity of TGF-β1’s effects has been proposed [77,78]. This model suggests that there is a critical expression level of the TGF-β1 receptor that determines the specific TGF-β1 responses of a cell [77,78,79]. In 2009, Rojas et al. provide direct evidence that the regulation of the expression level of TGFBR2 can affect the specificity of the TGF-β1 response [77]. They have shown that the activation and the intensity of the activation of the Smad and non-Smad signaling pathways can be modulated by the expression level of TGFBR2. The differences in activation of the Smad and MAPK-ERK pathways at different TGFBR2 levels suggests that not only can the level of TGFBR2 determine the intensity of pathway activation but also which pathways are activated [77,80]. The differential activation of MAPK-ERK and Smad pathways suggests that at low levels of TGF-β1 receptor activation the non-Smad signaling pathways predominate over the Smad signaling pathways in determining the effect of TGF-β1 on the cell [77].

In summary, the male and female sex hormones seem to influence the TGF-β1/TGF-β receptor axis by different mechanisms: estradiol rather via TGFBR1 and DHT via TGFBR2.

## 4. Conclusions

This report documents marked sex differences in diabetes- and TGF-β1-induced expression patterns in renal tissue. In addition, the TGF-β system is differentially affected by the sex hormones DHT and β-estradiol. Our data suggest that both genetic repertoire and sex hormones can determine sex differences in DN. Importantly, hormones and sex in combination with the TGF-β1 dose determine whether TGF-β1 exerts pro- or antifibrotic effects.

It is intriguing that our experiments did not provide evidence for the commonly invoked protective effects of estradiol. Further, in our experiments, none of the two hormones featured a dominant role in the pathophysiology of DM- and/or TGF-β1-induced renal damage. Indeed, each hormone exerted both stimulating and inhibitory effects on TGF-β signaling, depending on sex and/or TGF-β1 dosage. The data shown here therefore do not provide a conclusive answer to the question of whether a particular sex or sex hormone is a risk factor in the development or progression of DN. However, with a holistic approach, this study for the first time reveals complex gender differences in diabetic kidney disease on a molecular basis. Based on our findings, we propose that a combination of diabetes progress, sex, hormones and TGF-β1 levels best describes the sex effect, rather than gender alone. These findings should help to rationalize previous controversial literature data and pave the way for the design of sex-tailored therapies against DN modulating TGF-β1/BMP7 and androgen-to-estradiol ratios.

## Figures and Tables

**Figure 1 cells-09-02236-f001:**
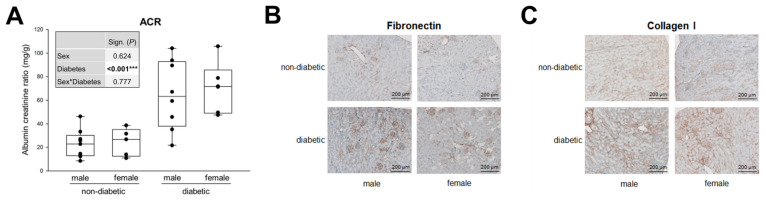
Microalbuminuria and TIF in diabetic mice. (**A**) Urinary albumin-to-creatinine ratio (ACR) from 24 h urine. Each point on the graph represents an individual mouse (*n* = 9/5/8/6). The significance, shown in the table (graph insert) was assessed using two-way ANOVA with sex (male or female) and diabetes (non-diabetic or diabetic) as the two factors. Sex, Diabetes: main effect of sex/diabetes, respectively. Sex*Diabetes: interaction (determines whether the one main effect depends on the level of the other main effect). (**B**) Fibronectin. Representative images of fibronectin immunohistochemistry (magnification: 100×). (**C**) Collagen I. Representative images of collagen type I immunohistochemistry (magnification: 100×). *** *P* ≤ 0.001.

**Figure 2 cells-09-02236-f002:**
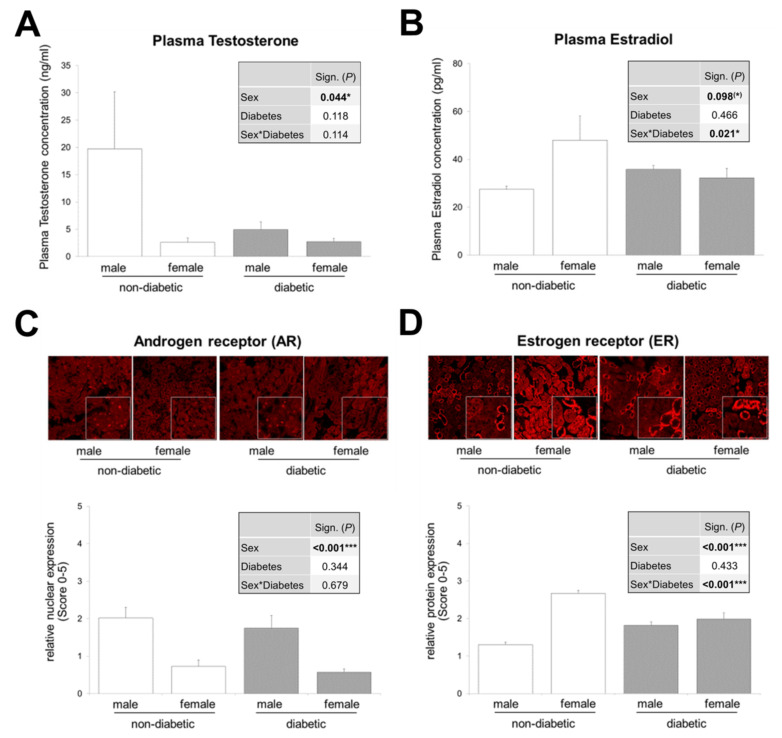
Sex hormone levels and expression of sex hormone receptors in both sexes of non-diabetic and diabetic mice. (**A**) Plasma testosterone (*n* = 5/3/10/8). (**B**) Plasma estradiol (*n* = 7/7/11/7). (**C**) Representative images of androgen receptor (AR) immunofluorescence and quantification of nuclear expression (graph) (*n* = 10/7/8/5 animals with 10 non-overlapping kidney areas). (**D**) Representative images of estrogen receptor (ER) immunofluorescence and quantification of expression (graph) (*n* = 8/7/9/6 animals with 10 non-overlapping kidney areas). (**A**–**D**) The significances, shown in the tables (graph inserts), were assessed using two-way ANOVA with sex (male or female) and diabetes (non-diabetic or diabetic) as the two factors. Sex, Diabetes: main effect of sex/diabetes, respectively. Sex*Diabetes: interaction (determines whether the one main effect depends on the level of the other main effect). Magnification of immunofluorescence (200×). * *P* ≤ 0.05; *** *P* ≤ 0.001.

**Figure 3 cells-09-02236-f003:**
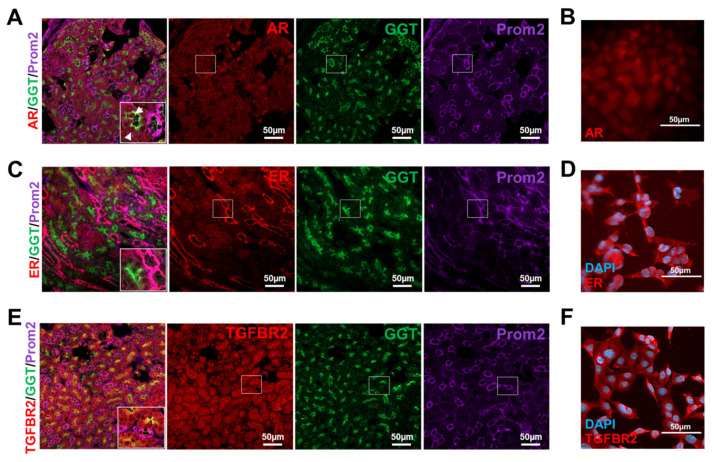
Expression of receptors for sex hormones and TGF-β1 in renal cortical tissue and TKPTS cells. (**A**) Androgen receptor (AR) immunofluorescence (red) of kidney sections with co-staining of marker for proximal tubular cells GGT (green) and marker for distal tubules Prom2 (purple). Arrows: AR and GGT co-stained cells. Magnification 200×. (**B**) Androgen receptor (AR) immunofluorescence (red) of TKPTS cells (magnification 400×). (**C**) Estrogen receptor (ER) immunofluorescence (red) of kidney sections with co-staining of marker for proximal tubular cells GGT (green) and marker for distal tubules Prom2 (purple). Magnification 200×. (**D**) Estrogen receptor (ER) immunofluorescence (red) of TKPTS cells, co-stained with DAPI (blue) (magnification 400×). (**E**) TGF-β receptor 2 (TGFBR2) immunofluorescence (red) of kidney sections with co-staining of marker for proximal tubular cells GGT (green) and marker for distal tubules Prom2 (purple). Magnification 200×. (**F**) TGF-β receptor 2 (TGFBR2) immunofluorescence (red) of TKPTS cells, co-stained with DAPI (blue) (magnification 400×). DAPI: 4’,6-diamidino-2-phenylindole; GGT: gamma-glutamyl transferase; Prom2: prominin 2.

**Figure 4 cells-09-02236-f004:**
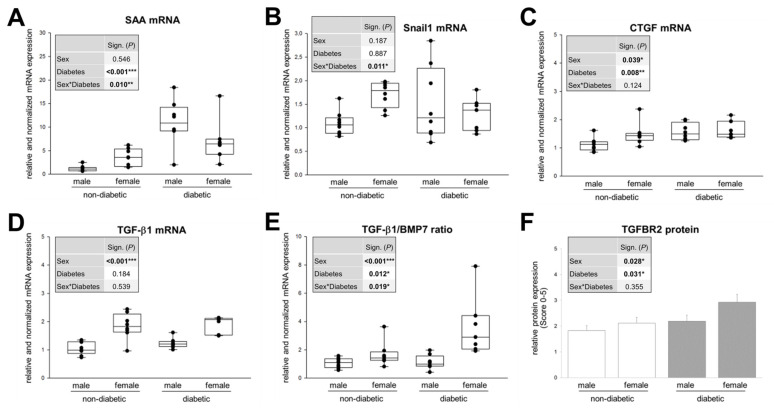
Expression of molecular markers of TIF in kidney cortex of non-diabetic and diabetic mice. (**A**) Real-time PCR analysis of SAA (serum amyloid A) mRNA. (**B**) Real-time PCR analysis of Snail1 mRNA. (**C**) Real-time PCR analysis of CTGF (connective tissue growth factor) mRNA. (**D**) Real-time PCR analysis of TGF-β1 mRNA. (**E**) Ratio of real-time PCR analysis of TGF-β1 and BMP7 mRNA. (**A**–**E**) Each point on the graph represents an individual mouse (*n* = 10/8/8/7). (**F**) Semi-quantitative analysis of the immunohistochemical staining of TGF-β1 receptor 2 (TGFBR2) (*n* = 9/7/9/7 animals with 10 non-overlapping kidney areas). (**A–F**) The significances, shown in the table insert, were assessed using two-way ANOVA with sex (male or female) and diabetes (non-diabetic or diabetic) as the two factors. Sex, Diabetes: main effect of sex/diabetes, respectively. Sex*Diabetes: interaction (determines whether the one main effect depends on the level of the other main effect). * *P* ≤ 0.05; ** *P* ≤ 0.01; *** *P* ≤ 0.001.

**Figure 5 cells-09-02236-f005:**
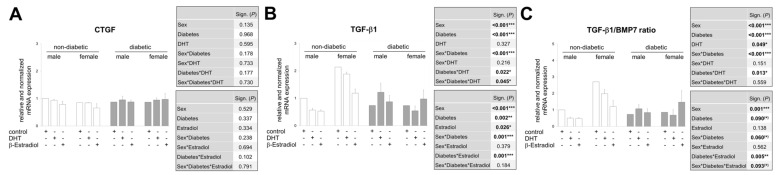
Real-time PCR analysis of mRNA expression in pieces of renal cortex treated in an ex vivo approach. (**A**–**C**) Treatment of cortical pieces from non-diabetic (3 males and 3 females) and diabetic (3 males and 3 females) mouse kidneys with control (DMSO) or sex hormones (DHT/β-estradiol) for 24 h. (**A**) CTGF mRNA analysis. (**B**) TGF-β1 mRNA analysis. (**C**) Ratio from TGF-β1 and BMP7 mRNA analysis. (**A**–**C**) The significances, shown in the upper table, were assessed using three-way ANOVA with sex (male or female), diabetes (non-diabetic or diabetic) and DHT as the three factors. The significances, shown in the lower table, were assessed using three-way ANOVA with sex (male or female), diabetes (non-diabetic or diabetic) and estradiol as the three factors. Sex, Diabetes, DHT, Estradiol: main effect of sex/diabetes/DHT/Estradiol, respectively. Sex*Diabetes, Sex*DHT, Diabetes*DHT, Sex*Estradiol, Diabetes*Estradiol: interaction (determines whether the one main effect depends on the level of the other main effect). Sex*Diabetes*DHT, Sex*Diabetes*Estradiol: interaction (determines whether the third factor modifies the interaction between the other two factors). * *P* ≤ 0.05; ** *P* ≤ 0.01; *** *P* ≤ 0.001.

**Figure 6 cells-09-02236-f006:**
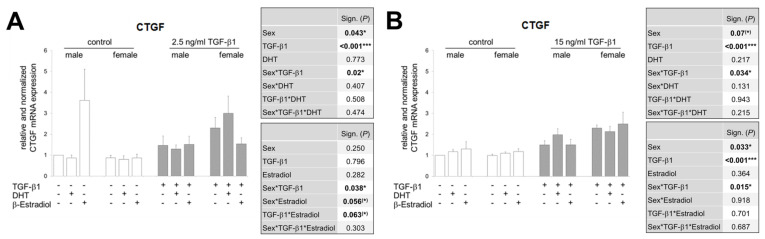
Effect of TGF-β1 stimulation on CTGF mRNA expression in renal tissues. (**A**) CTGF mRNA analysis. Both kidneys of each of female/male (*n* = 5 each) wildtype mice were minced pairwise, put into culture under control or 2.5 ng/mL TGF-β1 and with or without sex hormones for 24 h, as indicated. (**B**) CTGF mRNA analysis. Both kidneys of each of female/male (*n* = 6 each) wildtype mice were minced pairwise, put into culture under control or 15 ng/mL TGF-β1 and with or without sex hormones for 24 h, as indicated. (**A**,**B**) The significances, shown in the upper table, were assessed using three-way ANOVA with sex (male or female), TGF-β1 (control or 2.5 or 15 ng/mL) and DHT as the three factors. The significances, shown in the lower table, were assessed using three-way ANOVA with sex (male or female), TGF-β1 (control or 2.5 or 15 ng/mL) and estradiol as the three factors. Sex, TGF-β1, DHT, Estradiol: main effect of sex/TGF-β1/DHT/Estradiol, respectively. Sex*TGF-β1, Sex*DHT, TGF-β1*DHT, Sex*Estradiol, TGF-β1*Estradiol: interaction (determines whether the one main effect depends on the level of the other main effect). Sex*TGF-β1*DHT, Sex*TGF-β1*Estradiol: interaction (determines whether the third factor modifies the interaction between the other two factors). * *P* ≤ 0.05; *** *P* ≤ 0.001.

**Figure 7 cells-09-02236-f007:**
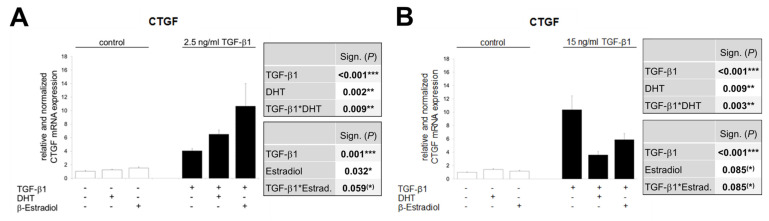
Real-time PCR analysis of mRNA expression in TGF-β1-treated TKPTS cells. (**A**) Cells were stimulated for 24 h with control or 2.5 ng/mL TGF-β1 and with or without sex hormones (DHT/β-estradiol), as indicated. CTGF mRNA levels were determined from duplicate samples in three independent experiments (yielding a total of *n* = 6 sample points). (**B**) Cells were stimulated for 24 h with control or 15 ng/mL TGF-β1 and with or without sex hormones (DHT/β-estradiol), as indicated. CTGF mRNA levels were determined from triplicate samples in four independent experiments (yielding a total of *n* = 12 sample points). (**A**,**B**) The significances, shown in the upper table, were assessed using two-way ANOVA with TGF-β1 (control or 2.5 or 15 ng/mL) and DHT as the two factors. The significances, shown in the lower table, were assessed using two-way ANOVA with TGF-β1 (control or 2.5 or 15 ng/mL) and estradiol as the two factors. TGF-β1, DHT, Estradiol: main effect of TGF-β1/DHT/Estradiol, respectively. TGF-β1*DHT, TGF-β1*Estradiol: interaction (determines whether the one main effect depends on the level of the other main effect). * *P* ≤ 0.05; ** *P* ≤ 0.01; *** *P* ≤ 0.001.

**Figure 8 cells-09-02236-f008:**
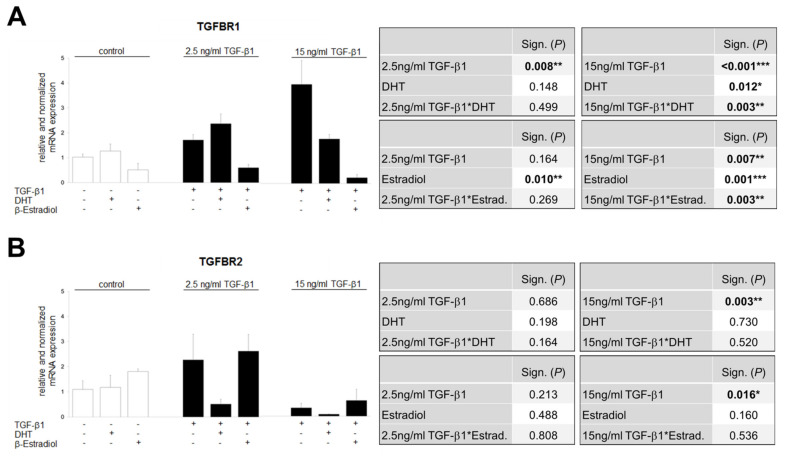
Effect of TGF-β1 on expression of TGF-β1 receptor 1 and 2 in TKPTS cells. (**A**) TGFBR1 mRNA analysis. Cells were stimulated for 24 h with TGF-β1 (control or 2.5 or 15 ng/mL) and with or without sex hormones (DHT/β-estradiol), as indicated (*n* = 4/6/2/6/6/2/2/5/3). (**B**) TGFBR2 mRNA analysis. Cells were stimulated for 24 h with TGF-β1 (control or 2.5 or 15 ng/mL) and with or without sex hormones (DHT/β-estradiol), as indicated (*n* = 3/3/2/3/3/3/4/5/3). (**A**,**B**) The significances, shown in the upper tables, were assessed using two-way ANOVA with TGF-β1 (control or 2.5 (left table) or 15 ng/mL (right table)) and DHT as the two factors. The significances, shown in the lower tables, were assessed using two-way ANOVA with TGF-β1 (control or 2.5 (left table) or 15 ng/mL (right table)) and estradiol as the two factors. TGF-β1, DHT, Estradiol: main effect of TGF-β1/DHT/Estradiol, respectively. TGF-β1*DHT, TGF-β1*Estradiol: interaction (determines whether the one main effect depends on the level of the other main effect). * *P* ≤ 0.05; ** *P* ≤ 0.01; *** *P* ≤ 0.001.

**Table 1 cells-09-02236-t001:** List of primer pairs and annealing temperatures.

Gene	Sense Primers	Antisense Primers	T_ann._
*HPRT1*	5′-TGGATACAGGCCAGACTTTGTT-3′	5′-CAGATTCAACTTGCGCTCATC-3′	59 °C [36]
*CTGF*	5′-TGCTGTGCAGGTGATAAAGC-3′	5′-AAGGCCATTTGTTCACCAAC-3′	58 °C [37]
*TGF-β1*	5′-AAGGGCTACCATGCCAACTT-3′	5′-CGGGTTGTGTTGGTTGTAGA-3′	62 °C [38]
*BMP7*	5′-GATTTCAGCCTGGACAACGAG-3′	5′-GGGCAACCCTAAGATGGACAG-3′	58 °C [39]
*SAA*	5′-TCATTTGTTCACGAGGCTTTC-3′	5′-ATGGTGTCCTCATGTCCTCTG-3′	59 °C [40]
*Snail1*	5′-GCGGAAGATCTTCAACTGCAAATATTGTAA-3′	5′-GCAGTGGGAGCAGGAGAATGGCTTCTCAC-3′	54 °C [41]
*TGFBR1*	5′-GCTCTAGATTTCTGCCACCTCTGTAC-3′	5′-GCGAATTCGACAGTGCGGTTATGGCA-3′	62 °C [42]
*TGFBR2*	5′-GCAGGCATCAGGACCTCAGTTTGATCC-3′	5′-AGAGTGAAGCCGTGGTAGGTGAGCTTG-3′	62 °C [42]

*HPRT1* = hypoxanthine phosphoribosyl transferase 1, *CTGF* = connective tissue growth factor, *TGF-β1* = transforming growth factor-β1, *BMP7* = bone morphogenetic protein 7, *SAA* = serum amyloid A, *Snail1* = snail family transcriptional repressor 1, and *TGFBR1/2* = TGF-β receptor 1/2.

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
