# Peer review of "Sex Differences in Diabetes- and TGF-β1-Induced Renal Damage"

_cells, 2020, doi:10.3390/cells9102236_

Round 1

Reviewer 1 Report

This is a generally well-written and research article that attempts to provide a definitive solution to apparent discrepancies in the literature regarding sex differences in diabetic nephropathy as related to TGF-beta induced renal damage. As a major complication of diabetes, the subject is widely researched and any additional insight that provides clarity is very welcome. I have a few very minor comments for the authors to consider:

Could the authors expand on the statement that accumulating evidence suggests that diabetes is associated with an imbalance in sex hormones? Please see line 92. References are provided, but more detailed would improve context for the current article.

Line 510-514. It is surprising that the authors do not resolve an increase in TGF-beta1. The explanation for a given time point may be true, but this needs evidenced and should only be cited if measurements at other time points were actually made. Was this the case, please add. If not, please revise the text.

Line 564-565. Can this statement be substantiated? If not, please omit or provide a more detailed explanation.

Very Minor

Line 98 does not need to be a new paragraph.

Line 110. Sentence should begin with “The mouse model….”

Line 310, please change to “If we only focus on non-diabetic animals, as expected we find that plasma testosterone…..”

Line 319. Please do not describe humans with diabetes as “diabetics”. Many patients rightly do not wish to be defined by their disease and it would be better to write “ males with diabetes”. This is in line with the advice from many diabetes-focused medical charities.

Figure 3. Why is the scale 20μm in panel B, but 50μm in panel D and F. Do these scales also correspond to panels A, C and E? Also please size B, D and F to match A, C and E.

Line 455, remove comma after “is”.

Line 503, please correct sentence structure after “detectable. lead to….”

Line 505 is not a new paragraph

Line 530, please add “it” to “with the data shown here “it” can…..”

Lines 623 and 628, please write the word Figure in bold.

Line 627 please change, “ DHT has minimal or inhibitory effects….”

Figure 5, panels A and B. Please alter the axis from 0-5 to 0-3. This will expand the data and make it easier for the reader to resolve changes.

General remark: There is some very early literature describing phenol red in media as being mildly estrogenic. For in vitro studies was the media phenol-red free?

Author Response

Reviewer: This is a generally well-written and research article that attempts to provide a definitive solution to apparent discrepancies in the literature regarding sex differences in diabetic nephropathy as related to TGF-beta induced renal damage. As a major complication of diabetes, the subject is widely researched and any additional insight that provides clarity is very welcome. I have a few very minor comments for the authors to consider:

Authors: Thank you. We have revised the manuscript according to your comments. We marked all changes in yellow to simplify the reviewing process.

Reviewer: Could the authors expand on the statement that accumulating evidence suggests that diabetes is associated with an imbalance in sex hormones? Please see line 92. References are provided, but more detailed would improve context for the current article.

Authors: We have included more details from the literature. Please see lines 88-94.

Reviewer: Line 510-514. It is surprising that the authors do not resolve an increase in TGF-beta1. The explanation for a given time point may be true, but this needs evidenced and should only be cited if measurements at other time points were actually made. Was this the case, please add. If not, please revise the text.

Authors: This is a good point. Indeed, no TGF-ß1 measurements at other time points were made. We have rephrased the text accordingly. Please see lines 495-501.  

Reviewer: Line 564-565. Can this statement be substantiated? If not, please omit or provide a more detailed explanation.

Authors: This critique is correct and we have removed this statement. Please see line 547.

Very Minor

Reviewer: Line 98 does not need to be a new paragraph.

Authors: We have changed it as proposed. Please see line 97.

Reviewer: Line 110. Sentence should begin with “The mouse model….”

Authors: We have changed it as proposed. Please see line 108.

Reviewer: Line 310, please change to “If we only focus on non-diabetic animals, as expected we find that plasma testosterone…..”

Authors: We have changed the text as proposed. Please see lines 303-304.

Reviewer: Line 319. Please do not describe humans with diabetes as “diabetics”. Many patients rightly do not wish to be defined by their disease and it would be better to write “ males with diabetes”. This is in line with the advice from many diabetes-focused medical charities.

Authors: Absolutely correct. Thank you for the advice. We are sorry about this expression. We have changed it as proposed. Please see line 311.

Reviewer: Figure 3. Why is the scale 20μm in panel B, but 50μm in panel D and F. Do these scales also correspond to panels A, C and E? Also please size B, D and F to match A, C and E.

Authors: Thanks for raising this point. Panel B was captured with another magnification than panels D and F (630x vs. 400x). Indeed, the reviewer is right in that this is somewhat confusing as shown. We have replaced the image B in Figure 3 with one with 400x magnification. Furthermore, the size bars were inserted into the images A, C, E to match B, D and F.

Reviewer: Line 455, remove comma after “is”.

Authors: We have changed it as proposed. Please see line 443.

Reviewer: Line 503, please correct sentence structure after “detectable. lead to….”

Authors: We have corrected the sentence accordingly. Please see line 490.

Reviewer: Line 505 is not a new paragraph

Authors: We have changed it as proposed. Please see line 491.

Reviewer: Line 530, please add “it” to “with the data shown here “it” can…..”

Authors: We have changed it as proposed. Please see line 516.

Reviewer: Lines 623 and 628, please write the word Figure in bold.

Authors: We have changed it as proposed. Please see lines 603 and 608.

Reviewer: Line 627 please change, “ DHT has minimal or inhibitory effects….”

Authors: We have changed it as proposed. Please see line 617.

Reviewer: Figure 5, panels A and B. Please alter the axis from 0-5 to 0-3. This will expand the data and make it easier for the reader to resolve changes.

Authors: Thank you for this good advice. We have changed the axis as suggested.

Reviewer: General remark: There is some very early literature describing phenol red in media as being mildly estrogenic. For in vitro studies was the media phenol-red free?

Authors: Many thanks for this hint. It is absolutely correct that phenol red may exhibit in cell culture very weak estrogen effects ( PNAS 83: 2496-2500; 1986 Berthois et al. und Mol Cell Endo 57: 169-178; 1988 Welshons et al). Consequently we performed the pivotal ex vivo experiments testing hormonal effects without phenol red (see figure 5). The media for in vitro studies, however, did contain phenol-red. Having said this, we never observed effects of phenol red in our experiments. Therefore we do not think phenol red influenced our results.

Reviewer 2 Report

This study demonstrated the influence of sex hormones in affecting the diabetic nephropathy in dbdb mice. This study is interesting, and the scientific value is good. However, several comments are suggested: 

  1. The manuscript is too long ! And I think it is a little difficult to read. The authors may consider to shorten the manuscript and to give the readers refined words. 
  2. Only mRNA expression or IHC were presented in the manuscript. The authors may consider present some data with western blot analysis to convince the readers. 
  3. In the part of ex vivo studies, the authors only evaluate the changes of fibrotic cytokines after DHA or estradiol stimulation. The authors should consider to provide further information about the expression of fibrosis markers (mRNA or WB, such as collagen IV, fibronectin) after the changes of upstream fibrotic cytokines. 
  4. The conclusion was not refined and seemed as no conclusion ?. The authors may consider to give a summary figure for the manuscript. 

Author Response

Reviewer: This study demonstrated the influence of sex hormones in affecting the diabetic nephropathy in dbdb mice. This study is interesting, and the scientific value is good. However, several comments are suggested: 

Authors: Thank you for your suggestion. We marked all changes in green to simplify the reviewing process.

Reviewer: The manuscript is too long ! And I think it is a little difficult to read. The authors may consider to shorten the manuscript and to give the readers refined words. 

Authors: Thank you for your suggestion. We have shortened and revised the manuscript according to your suggestion, trying not to lose rigor and precision. To simplify reviewing, we marked all changes in the text as follows: For changes within a sentence, we marked the first word in green and for deletions we marked the last word of the preceding sentence in green. 

Reviewer: Only mRNA expression or IHC were presented in the manuscript. The authors may consider present some data with western blot analysis to convince the readers.

Authors: Thank you for your comment. While in principle we agree that western Blotting could substantiate our results, in our experience these experiments are technically difficult to perform with (fibrotic) mouse kidneys (e.g. they require pooling of various kidney, complicate homogenization, etc.). We have found that protein analysis via IHC is more reliable, reproducible and as such more convincing. Moreover IHC also informs on the cell types expressing particular proteins, which is not possible in total protein lysate. Considering this, and since we do not see major gender-dependent differences at the protein level in fibrosis marker expression, we propose to avoid western blotting and rely on the IHC data as presented.

Reviewer: In the part of ex vivo studies, the authors only evaluate the changes of fibrotic cytokines after DHA or estradiol stimulation. The authors should consider to provide further information about the expression of fibrosis markers (mRNA or WB, such as collagen IV, fibronectin) after the changes of upstream fibrotic cytokines.

Authors:  These are good points. Indeed, we are planning to do this kind of time-line experiments in future. These are complex, multi-parameter and work-intensive experiments and thus we think they are beyond the scope of the present study.

Reviewer: The conclusion was not refined and seemed as no conclusion ?. The authors may consider to give a summary figure for the manuscript.

Authors: Thank you for raising this point. We have refined and streamlined the conclusion and tried to make it more punchy. We think that a figure would be too complicated to better explain the highly complex effects/relationships.

Round 2

Reviewer 2 Report

I think the authors have proper response to the my review suggestions. I think it is OK to accept this paper. Thanks !